# Fundamentals of Rhenium-188 Radiopharmaceutical Chemistry

**DOI:** 10.3390/molecules28031487

**Published:** 2023-02-03

**Authors:** Janke Kleynhans, Adriano Duatti, Cristina Bolzati

**Affiliations:** 1Radiopharmaceutical Research, Department of Pharmaceutical and Pharmacological Sciences, Katholieke Universiteit Leuven, 3000 Leuven, Belgium; 2Department of Chemical and Pharmaceutical Sciences, University of Ferrara, 44121 Ferrara, Italy; 3ICMATE-CNR, Corso Stati Uniti 4, 35127 Padua, Italy

**Keywords:** rhenium, transition metal chemistry, ligand field theory, frontier orbitals, nuclear medicine, ^188^W/^188^Re generator, perrhenate, targeted radionuclide therapy (TRNT), theranostic

## Abstract

The β^−^ emitter, rhenium-188 (^188^Re), has long been recognized as an attractive candidate for targeted cancer radionuclide therapy (TRNT). This transition metal shares chemical similarities with its congener element technetium, whose nuclear isomer technetium-99m (^99m^Tc) is the current workhorse of diagnostic nuclear medicine. The differences between these two elements have a significant impact on the radiolabelling methods and should always receive critical attention. This review aims to highlight what needs to be considered to design a successful radiopharmaceutical incorporating ^118^Re. Some of the most effective strategies for preparing therapeutic radiopharmaceuticals with ^188^Re are illustrated and rationalized using the concept of the inorganic functional group (core) and a simple ligand field theoretical model combined with a qualitative definition of frontiers orbitals. Of special interest are the Re(V) oxo and Re(V) nitrido functional groups. Suitable ligands for binding to these cores are discussed, successful clinical applications are summarized, and a prediction of viable future applications is presented. Rhenium-188 decays through the emission of a high energy beta particle (2.12 MeV max energy) and a half-life of 16.9 h. An ideal biological target would therefore be a high-capacity target site (transporters, potential gradients, tumour microenvironment) with less emphasis on saturable targets such as overexpressed receptors on smaller metastases.

## 1. Introduction

At the end of the last century, the radioisotope rhenium-188 (^188^Re) was among the first radionuclides to be proposed for the development of therapeutic radiopharmaceuticals [1]. There were several reasons to believe that ^188^Re was an appropriate choice for medical applications. The nuclear properties of ^188^Re are extremely interesting. It decays through the emission of β^−^ particles with mean energy E_mean_ = 763 keV and maximum energy E_max_ = 2.12 MeV which is advantageous for the delivery of cytotoxic radiation to the tumour. The emission of a γ photon, with energy E_γ_ = 155.04 keV (15.05%), is also associated with the β^−^ decay. This isomeric transition is almost ideal for applying single photon emission tomography (SPECT) and outperforms emissions from other β^−^ emitting radionuclides in terms of resolution for dosimetry purposes [2]. The relatively short half-life of 16.98 h when combined with the very high specific activity of this radionuclide can become an additional advantage for some specific clinical applications as discussed later in this article [3,4].

An attractive feature that initially contributed to stimulating the interest in ^188^Re was the similarity between the chemical properties of the two congener elements rhenium and technetium [5,6,7]. The reasoning was that ^188^Re radiopharmaceuticals could be produced simply by replacing the radionuclide ^99m^Tc with the radionuclide ^188^Re in the well-established reaction scheme used for the preparation of ^99m^Tc radiopharmaceuticals. In the periodic table, technetium and rhenium belong to Group 7, and are categorized in the second and third row of the transition metal series, respectively. A variety of radiopharmaceuticals labelled with the metastable γ emitting nuclear isomer ^99m^Tc for the diagnosis of many pathological conditions has been investigated [1,5]. This was accomplished by the development of efficient chemical methods for the synthesis of ^99m^Tc complexes in physiological solutions and in sub-millimolar concentrations (also called ‘no-carrier-added level’). A prerequisite for clinical translation was safety and stability during intravenous administration in vivo. The development of highly efficient labelling methods stemmed from the richness of the chemical properties of the element technetium. This unique characteristic of Group 7 transition metals allowed the production of a multitude of technetium complexes characterized by a wide variety of molecular structures. The reader is referred to in-depth reviews on the topic of ^99m^Tc chemistry [1,6,8].

Similarly, it was argued that the same structural richness could be observed for ^188^Re radiopharmaceuticals which ultimately could be used as therapeutic agents with almost no need for optimization of labelling methods. This chemical parallelism between the two radioisotopes was interpreted as one of the earliest examples of ‘theranostic pair’ (although the term ‘matched pair’ was originally used), a ubiquitous concept in current radiopharmaceutical research [9]. Nature is not always respectful of the elegant symmetries described in scientists’ theoretical models and usually deviates from the expected results. In the example of the congener elements rhenium and technetium, the chemical similarities between the two elements are only superficial, and this diversity becomes more pronounced when studying their chemical reactivity at the no-carrier-added level as implemented in radiopharmaceutical production. This forced the researchers to abandon the hope that ^188^Re radiopharmaceuticals could be produced simply by replacing the nuclide ^99m^Tc with the nuclide ^188^Re in the well-established reactions utilized for the preparation of ^99m^Tc radiopharmaceuticals. Commonly, when this approach was naively attempted, the production of the corresponding ^188^Re radiopharmaceuticals failed dramatically. Therefore, it became clear that it was necessary to develop specific labelling methods for preparing ^188^Re radiopharmaceuticals and that the methods used for the preparation of ^99m^Tc radiopharmaceuticals were simply not superimposable [9,10].

The introduction of regulatory constraints in the radiopharmacy based on the principles of Good Manufacturing Practice (GMP) strongly favour simpler and less demanding reaction routes for the preparation of radiopharmaceuticals. Therefore, the lack of more efficient procedures to produce different types of ^188^Re radiopharmaceuticals, has significantly hampered the extension of this category of therapeutic agents. This review aims to discuss and analyse from a fundamental point of view the chemical problems that have hitherto prevented the development and clinical application of ^188^Re radiopharmaceuticals on the scale of other therapeutic radionuclides. An alternative point of view is offered here that questions the prevailing opinion that radiopharmaceuticals containing α-emitters (e.g., actinium-225) or long-half-life, low-energy β^−^-emitters (e.g., lutetium-177) [11], constitute the best option for an effective targeted radionuclide therapy (TRNT).

In particular, the present review was designed to provide a deeper insight into the fundamental chemistry of ^188^Re radiopharmaceuticals, a topic that was not extensively covered in the current radiopharmaceutical literature. A basic description of the chemical bonding in ^188^Re radiopharmaceuticals containing the Re oxo ([Re≡O]^3+^) and Re nitrido ([Re≡N]^2+^) cores has been devised following a qualitative and heuristic approach inspired by the analogy with organic functional groups and by the theory of frontier molecular orbitals (Hoffman’s theory), which was utilized here to discuss the chemical reactivity of these inorganic moieties. This qualitative and pictorial representation was preferred to more rigorous and modern theoretical models (e.g., DFT = density functional theory) because it was considered more accessible to Radiopharmacists and non-experts in sophisticated quantum theoretical descriptions. For the reasons given above, an in-depth discussion of medical applications of ^188^Re radiopharmaceuticals was considered out of the scope of the present review and only a few examples have been briefly mentioned.

## 2. The Reduction of the Perrhenate Anion [^188^Re][ReO_4_]^−^

The radioisotope ^188^Re is produced from the weak β^−^ decay of the parent nuclide tungsten-188 (^188^W). The parent nuclide is in turn obtained by double neutron capture of a ^186^W target after neutron irradiation in a high-flux nuclear reactor according to the reaction pathway ^186^W + 2*n* → ^188^W → ^188^Re + β^−^ + $\bar{\nu }$ (where *n* = neutron and $\bar{\nu }$ = electron antineutrino). The produced ^188^W can then be loaded onto a generator system to afford a long-term source of no-carrier added ^188^Re. This makes the use of ^188^Re in therapeutic applications even more fortuitous [12,13,14].

The analogous ^188^W/^188^Re generator follows the same design strategy utilized for the development of the ^99^Mo/^99m^Tc generator where the long-lived ^188^W parent nuclide (69.4 days) is absorbed on an alumina column. This system can be periodically eluted with a sterile saline solution (0.9% NaCl) that allows the selective separation of ^188^Re, in the chemical form of sodium perrhenate ([^188^Re]ReO_4_Na), in very high specific activity [12,13,14].

Similarly to the pertechnetate anion, [^99m^Tc][TcO_4_]^−^, the perrhenate anion, [^188^Re][ReO_4_]^−^, is the universal starting material for the preparation of ^188^Re radiopharmaceuticals. The two tetraoxo anions, [MO_4_]^−^ (M = Re, Tc), share the same molecular structure where the metal is bound to four oxygen atoms in a tetrahedral arrangement. Consequently, the preparation of new ^188^Re compounds necessarily requires the removal of at least one, or ultimately, all four of the oxygen atoms from the perrhenate anion. This process can be accomplished through the reaction of [^188^Re][ReO_4_]^−^ with an appropriate chemical species (:A) able to accept oxygen atoms. A schematic, step-by-step representation of this reaction is reported in Figure 1.

Traditionally, the overall reaction scheme depicted in Figure 1 is called the reduction of the [^188^Re][ReO_4_]^−^ anion and this definition originates from a different description of the oxo-transfer reaction based on the concept of ‘oxidation state’. The attribution of a positive or negative number (usually an integer number) to each atom of a particular element when bound to other atoms inside a molecule is a very old procedure which, nonetheless, still provides a very convenient method to classify the various chemical properties of a single element. The oxidation states of an element are purely formal numbers as they cannot be derived from a more fundamental theory. On the contrary, they are calculated following an abstract set of rules (that can be conveniently modified or extended when necessary). To simplify here, considering that oxygen and hydrogen play a dominant role on a planet covered by water, the oxidation states −2 and +1 are arbitrary attributed to these two elements, respectively, when they form a molecule. The other key assumption necessary to calculate the oxidation states is that the sum of all oxidation states of all elements in a molecule should be equal to the overall net charge of that molecule. Based on these simple rules, it is straightforward to calculate the oxidation state of the rhenium atom in the [^188^Re][ReO_4_]^−^ anion. Since this molecular anion carries an overall negative charge of –1, and the rhenium metal is bound to four oxygen atoms, the metallic oxidation state, *x*, should satisfy the following equation: *x* + [4 × (−2)] = –1 resulting in *x* = +7. In the reaction diagram of Figure 1, assuming that the contribution to net charge of the molecule by the ligands L*^n^* (*n* = 1–4) is zero, it follows that the removal of each oxygen atom by the acceptor :A necessarily decreases the oxidation state of rhenium by a factor 2 as, in the final sum of the overall oxidation states, one oxygen atom is missed. This means that for each oxygen-atom transfer, the oxidation state of rhenium is progressively reduced from +7 to a lower oxidation state (a process called ‘reduction’). In conclusion, the oxygen-atom transfer reaction starting from [^188^Re][ReO_4_]^−^ can also be also viewed as a ‘reduction of the perrhenate anion’. Obviously, when there exists a non-zero contribution by the ligand L*^n^* to the overall sum of the oxidation states, the final oxidation state of rhenium should be calculated accordingly.

It is worth noting here that it is not chemically correct to define a metal in a +*n* positive oxidation state as ‘*n*-valent’ (heptavalent, pentavalent, tetravalent, trivalent, etc.). This usage is quite common in radiopharmaceutical chemistry and its origin dates to some archaic conventions where positive oxidation states are represented by Roman numerals (for example, +7 is also written as VII). This is incorrect, because the definition of ‘valency’ does not refer to the value of the oxidation state but, rather, to the number of chemical bonds an element X can form with hydrogen in their simpler binary compounds XH*_n_* (*n* = valency of the element X). A clear example of how conceptually wrong this convention is can be found in the hydride complex [ReH_9_]^2−^. In metal–hydride complexes, the conventional rule that hydrogen has an oxidation state +1 must be modified and this element is assigned an oxidation state −1. As a result, the rhenium atom in [ReH_9_]^2−^ should possess an oxidation state +7 although this metal is bonded to nine hydrogen atoms and, consequently, its valency is 9 [15,16].

The ligands L*^n^* play a critical role in the reactions illustrated in Figure 1 because they fundamentally contribute to stabilizing the metal oxidation state arising after the oxygen atom transfer. In summary, the production of a stable ^188^Re radiopharmaceutical always requires the following key ingredients:

(i)generator-eluted [^188^Re][ReO_4_]^−^ in high-specific activity,(ii)an appropriate oxygen atom acceptor ‘:A’ (historically called ‘reducing agent’),(iii)appropriate ligands for stabilization of the metallic complex.

The reaction scheme reported in Figure 1 entirely parallels that employed for the preparation of ^99m^Tc radiopharmaceuticals where [^188^Re][ReO_4_]^−^ is replaced by the analogous tetraoxo anion [^99m^Tc][TcO_4_]^−^. However, a remarkable difference exists between these two radioelements that can be summarized as follows: when the same reaction scheme used for the preparation of a ^99m^Tc radiopharmaceutical is applied to the preparation of the analogue ^188^Re radiopharmaceutical, usually the observed radiochemical yield is dramatically lower than that observed with congener radioelement. It may even be that the reaction does not take place at all. This issue is commonly reported in the scientific literature as ‘the problem of the reduction of perrhenate’. To illustrate the origin of this conundrum, it is necessary to resort to a thermodynamic approach.

As discussed above, the preparation of a ^188^Re radiopharmaceutical can be also viewed as a redox reaction in which the oxidation state of metallic rhenium in [^188^Re][ReO_4_]^−^ is reduced by the action of some suitable reducing agent. The value of the standard reduction potential (*E°*) is the thermodynamic constant that gives a measure of the tendency for this reduction process to occur. It is well established that the value of *E°* for the reduction of [^188^Tc][TcO_4_]^−^ is always significantly higher than that for the same reaction carried out with [^188^Re][ReO_4_]^−^ (commonly, this difference is of the order of hundreds of millivolts). This indicates that the preparation of ^188^Re radiopharmaceuticals is far more difficult than that of the corresponding ^99m^Tc radiopharmaceuticals and a different chemical strategy needs to be applied to develop a viable method for the high-yield production ^188^Re radiopharmaceuticals.

## 3. The Expansion of the Coordination Sphere

The electrochemical potential for a redox reaction (Δ*E*) is regulated by the Nernst equation, which in turn depends on the difference between the standard reduction potential of the reductant and that of the oxidant (∆*E°*), the absolute temperature (*T*), and the logarithm of the concentration ratio (*Q*) between products and reactants:$$\left[\Delta E=\Delta E{}^{\circ}-\frac{RT}{nF}lnQ\right]$$
where *n* is the number of moles of electrons exchanged in the reaction and *F* is the Faraday constant). Since the impact of the logarithmic term on the value of Δ*E* is less than 10%, it is arduous to increase the yield of a redox process simply acting on the macroscopic variables such as temperature and concentration. Therefore, other chemical approaches should be employed to raise the radiochemical yield for the reduction of [^188^Re][ReO_4_]^−^. Some insight on a possible strategy can be obtained by considering the thermodynamic expression of the electrochemical potential:$$\Delta E=-\frac{\Delta G}{nF}$$
where Δ*G* is the variation of the free energy of the reaction and *n* is still the number of moles of electrons exchanged in the redox process and *F* the Faraday’s constant. In turn,
ΔG=ΔH−TΔS
where Δ*H* is the variation of enthalpy, or of the heat of reaction at constant pressure, and Δ*S* is the change in entropy. Obviously, for a given reaction, the formation of products is favoured when the value of Δ*G* is negative. A negative value of Δ*H* can be obtained only through the formation of more stable chemical bonds in product molecules, or by acting on the temperature. However, under normal conditions, these factors alone cannot bring about a significant change of the free energy, and consequently, of the electrochemical potential Δ*E*, necessary to increase the yield for the reduction of [^188^Re][ReO_4_]^−^.

The remaining factor left for modification in the expression of Δ*G* is the variation of entropy Δ*S*. The change in entropy should be largely positive to ensure the negativity of Δ*G*. In the language of statistical thermodynamics, an increase in entropy entails a concomitant increase of the degrees of freedom in going from reactants to products. Here, the initial system is [^188^Re][ReO_4_]^−^, which is a molecular ion having a tetrahedral geometry. An increase in entropy occurs only if the tetrahedral, four-coordination arrangement around the central rhenium atom can be expanded to achieve a higher coordination number, namely, five-, six, or higher. This expansion is necessarily linked to a corresponding change of the molecular geometry from tetrahedral to higher geometries (for example, square-pyramidal, octahedral, and dodecahedral).

Based on the above considerations, a possible strategy to increase the radiochemical yield of labelling reactions involving the [^188^Re][ReO_4_]^−^ anion is to first expand the coordination number of the rhenium atom, while preserving its oxidation state +7 (also written as VII), before carrying out the reduction reaction [17]. This can be accomplished by using suitable coordinating ligands able to form coordination complexes with rhenium in the oxidation state +7 (VII). Following this strategy, the oxygen transfer (reduction) reaction does not occur between tetrahedral Re(VII) perrhenate, but from a Re(VII) complex with an expanded coordination geometry around the metal. Hypothetically, this could lead to the expected, significant increase of the standard reduction potential and to a boost in the yield of production of ^188^Re radiopharmaceuticals. In the scientific literature, this effect is defined as the ‘expansion of the coordination sphere’ [17].

In principle, there exists a large class of ligands-forming complexes with rhenium in the oxidation state +7, but only one example of the application of this chemical strategy to the preparation of ^188^Re-radiopharmaceuticals has been described. In these studies, the conventional reducing agent Sn^2+^ was used as an oxygen atom acceptor for preparing the complex [^188^Re]Re(N)(DEDC)_2_ where DEDC = diethyldithiocarbamate (see the following sections) [18]. However, most importantly, the reduction of [^188^Re][ReO_4_]^−^ was conducted in the presence of the sodium salt of the oxalate ion, ([C_2_O_4_]^2−^, under acidic conditions (acetic acid). Oxalate is a classical ligand for coordinating metals in high oxidation states and, as expected, in this reaction, its role was to form an intermediate Re(VII) complex having a coordination arrangement higher than that of the starting tetrahedral ^188^Re-perrhenate anion (for instance, square pyramidal, octahedral) [19,20,21,22]. The expansion of the coordination sphere around the Re(VII) ion impacts on the entropic factor of the free energy of the oxygen transfer reaction and leads to a sharp increase in the value of the standard reduction potential [17]. Recently, this reaction was greatly improved by replacing sodium oxalate and acetic acid with simple oxalic acid, thereby significantly reducing the number of steps in the radiopharmaceutical synthesis and the potential radiation exposure of the operator [23,24].

## 4. Basic Rhenium Coordination Chemistry

In organic chemistry, the notion of ‘functional group’ is a very useful approach for describing the chemical properties of an organic molecule. Using this approach, the structure of an organic molecule is viewed as made up of different molecular pieces each corresponding to a particular grouping of atoms. For example, the group of atoms composed by a nitrogen atom and two hydrogen atoms (NH_2_) constitutes the amino functional group and its presence imparts characteristic structural properties and reactivity to the hosting molecule. A similar approach can be also applied to rhenium chemistry and, consequently, several ‘inorganic functional groups’ (also called ‘cores’ or ‘metallic fragments’) have been identified. Likewise with the organic counterparts, an inorganic functional group governs the chemical behaviour of a specific class of metallic complexes and determine their structural properties. Some of the most important rhenium functional groups are the oxygenated cores Re(VII) trioxo {[Re(O)_3_]^+^}, Re(V) dioxo {[Re(O)_2_]^+^}, and Re(V) oxo {[Re(O)]^3+^}, the nitrido core {[Re(V)N]^2+^}, the diazenido core {[Re(V)(NNR)]^2+^} (where R is a terminal substituent), and the tris-carbonyl core {[Re(CO)_3_]^+^} [6,8,25,26].

A reproducible procedure for the high-yield preparation of ^188^Re radiopharmaceuticals from generator-produced [^188^Re][ReO_4_]^−^ has been successfully achieved only with a few cores. Specifically, ^188^Re-radiopharmaceuticals bearing the Re(V) oxo and Re(V) nitrido cores have been produced almost exclusively at tracer level and evaluated in human studies [4,27]. Exceptions are the ^188^Re radiopharmaceuticals [^188^Re][Re(PhCS_3_)_2_(PhCS_2_)] (Ph = phenyl group), which has been introduced for the treatment of liver cancer [3,28] and incorporates the [Re^III^(S_2_)(S_3_)_2_] functional moiety, and the polymeric ^188^Re bisphosphonate agents of unknown structure used for bone-pain palliation [29,30]. Attempts to produce ^188^Re radiopharmaceuticals using other cores have been always plagued by poor radiochemical yields and low stability under physiological conditions. For this reason, in this review, the discussion of the basic chemistry of ^188^Re radiopharmaceuticals will be restricted to complexes containing these two Re(V) functional groups.

### Bonding in Rhenium Complexes

Before entering the discussion of rhenium coordination chemistry, it could be helpful to recall here a few elementary concepts on the type of chemical bonds that the rhenium atom can form with various ligands. The hydrogen-like model of multi-electron atoms holds that transition metals possess partially filled, *d*-type external atomic valence orbitals (AOs). These orbitals can overlap with appropriate orbitals on the ligand to form a coordination complex. In Figure 2a, the possible overlaps between *p*-type orbitals are illustrated. The resulting bonding orbitals are either parity symmetric (σ) or antisymmetric (π) with respect to the bond axis. Similarly, in Figure 2b, σ-type or π-type bonds are obtained by overlapping *p* orbitals with *d_xy_* (or equivalently, *d_xz_*, *d_yz_*, *d_z_*^2^, *d_x_*^2^*_–y_*^2^) orbitals. Conventionally, Lewis bases forming σ bonds with the metal are called σ donors, whereas Lewis acids forming π bonds with the metal are called π-acceptors.

Pursuing the analogy with organic functional groups, to obtain a qualitative picture of the chemical properties of rhenium complexes that contain a characteristic inorganic functional group, it is necessary to use a theoretical model to describe its electronic structure. Since it is not required to describe the whole coordination complex, but of a smaller molecular piece, the simplest and most intuitive approach is to use the fragment molecular orbital theory. In this theory, the molecular orbitals (MOs) of the functional group are obtained by combining the valence hydrogen-like AOs of the constituent atoms following symmetry considerations. It is helpful to recall here that the a priori knowledge of the geometry of a molecule is essential for obtaining the symmetry-adapted linear combination of atomic orbitals (SA-LCAO) that generate the MOs. Since this information cannot be theoretically inferred from any existing fundamental theory such as quantum mechanics, it can only be experimentally determined [31,32].

To briefly introduce the SA-LCAO procedure, we limit the discussion here to the two examples of the [ReN]^2+^ and [ReO]^3+^ cores, as these functional groups have been already successfully utilized for the production of ^188^Re radiopharmaceuticals so far.

Figure 3a illustrates a square pyramidal (*sp*) geometry with the associated coordinate system. In this geometry, the rhenium atom lies on the *sp* plane, and the heteroatom (O, N) occupies the apical position over that plane. A simplified, thus not quantitative, energy diagram describing the symmetry-adapted MOs and electronic distribution for the Re(V)(O) and Re(V)(N) functional groups is reported in Figure 3b. A brief description of the diagram could be of help to explain its significance and utility. On the left, the *d*-type AOs of the Re(V) metal (*d_xy_*, *d_xz_*, *d_yz_*, *d_z_*^2^, *d_x_*^2^*_–y_*^2^) are arranged according to their energies in a square pyramidal geometry. According to group theory, these AOs lose their degeneracy when the spherical symmetry of the free atom is constrained into the lower *sp* symmetry typical of Re(V) oxo and Re(V) nitrido complexes (a). Similarly, on the right of Figure 3b, the splitting of the three *p* orbitals of the heteroatom (O or N) in the *sp* geometry is also illustrated. The SA-LCAOs are formed through the linear combinations of AOs that transform into each other under the symmetry operations of an *sp* geometry. Parity-symmetric (σ and σ*) and parity-antisymmetric (π and π*) MOs are obtained, and these are represented at the centre of the diagram. As dictated by crystal field theory, the *d_x_*^2^*_–y_*^2^ orbital rises at very high energy because of its stronger repulsive interaction with the ligands positioned on the *x* and *y* axes as shown in the upper part of the diagram [31,32]. The *d_xy_* is left as non-bonding orbital (*n*). These MOs are called the frontier orbitals (FOs) of the metallic fragments Re(O) and Re(N).

Though at first sight it may appear rather abstract, the diagram in Figure 3 nicely explains many chemical features of the Re(X) (X = O, N) cores. The overall count of valence electrons for these cores is 8, but only 3 electron pairs are hosted in bonding orbitals. This means that the Re–X bond order is 3 and the X heteroatom is tethered to the metal through a triple bond, a result predicting a high electronic stability for these functional groups. A very useful piece of information that can be inferred from the diagram in Figure 4 is that the electron pairs lying in low-energy MOs are all involved in the Re(X) (X = O, N) multiple bonding and, thus, are not available to form bonds with other ligands. The highest occupied molecular orbital (HOMO) is a non-bonding (*n*) FO containing one electron pair. The lowest unoccupied molecular orbitals (LUMO) are the empty π* FOs (Figure 4). Hence, it is reasonable to predict that the Re(X) (X = O, N) fragment can behave as a Lewis acid and bind to ligands containing donor atoms with filled atomic orbitals such as [−O]^−^, [−S]^−^ and amino nitrogen [:NH_2_]. This interpretation is fully inspired by the analogy with the more familiar organic functional groups, such as the carbonyl and isocyanide groups that are known to behave as π acceptor Lewis acids.

The electrophilic character of the Re(X) (X = O, N) groups disfavours their interaction with π-acceptor ligands. To allow this interaction, it is necessary to modify the molecular geometry around the Re(X) core to induce a change in the distribution of FOs. For example, the [Re≡O]^3+^ monoxo core is converted into the *trans*-[O=Re=O]^+^ dioxo core when coordinated by bidentate π-acceptor diphosphines (R_2_P–Y–PR_2_, Y = organic bridge) and forms monocationic octahedral complexes [Re(O)_2_(PR_2_–Y–PR_2_)_2_]^+^. Similarly, the reaction of this core with monodentate π-acceptor tertiary phosphines, PR_3_ (where R = organic functional groups) leads the molecular structure to switch from square-pyramidal to six-coordinate, octahedral geometry with the formation of the mixed halogeno-phosphine complexes Re(O)Cl_3_(PR_3_)_2_] [1,6,7,8,22,25].

An analogous structural change is also observed with the [Re≡N]^2+^ functional group when coordinated by the tertiary phosphine π-acceptor ligand PPh_3_ (Ph = phenyl group). In this reaction, the square pyramidal Re(V) nitrido precursor [Re(N)Cl_4_]^−^ is converted to the five-coordinate complex [ReNCl_2_(PPh_3_)_2_]. Surprisingly, this complex exhibits a pseudo trigonal bipyramidal (*tbp*) geometry where the [Re≡N]^2+^ group is placed at the centre of the trigonal equatorial plane and the two phosphine ligands span the two *trans* positions on the axis perpendicular to this plane [33]. However, it is important to emphasize here that the attribution of a *D_3h_* symmetry to the complex [Re(N)Cl_2_(PPh_3_)_2_] (in group theory, the *tbp* geometry is mathematically represented by the point group *D_3h_*) could be considered rather arbitrary. Unfortunately, in coordination chemistry, it is arduous to find real examples of metallic complexes assuming an almost ideal geometry as commonly steric hindrance and weak interactions can always distort their structures. For instance, in the complex [Re(N)Cl_2_(PPh_3_)_2_], the bulky PPh_3_ substituents are partially repelled by the electron-rich Re≡N moiety. Consequently, the Re–P bonds are bent away from their axial positions and the ideal *D_3h_* symmetry is lowered to a *C_2v_* symmetry. However, the preference here for an ideal *tbp* (*D_3h_*) geometry was dictated by the purpose to interpret, through a simple FOs model, the tendency of π-acceptor ligands to span a reciprocal *trans* position in five-coordinate Re and Tc nitrido complexes containing mixed σ,π-donor and bidentate π-acceptor ligands. Remarkably, experimental evidence [34,35,36,37,38] showed that in bis-substituted Tc and Re nitrido complexes with bidentate phosphino-thiol ligands [35,36,37,38], the two π-acceptor phosphorous atoms always prefer to occupy the two axial positions of a *tbp* geometry. As discussed below and illustrated in Figure 5, the choice of a *D_3h_* symmetry allowed to draw a simple FOs diagram and, presumably, to find a more intuitive interpretation of these findings. Conversely, a *C_2v_* symmetry necessarily led to a more complex FOs diagram.

Figure 5 reports the MO diagram of the [Re≡N]^2+^ functional group in a *tbp* geometry. A remarkable feature of this energy diagram is that the symmetry-adapted FOs are separated in different sets corresponding to one axial, filled FO (π_axial_), and two empty equatorial FOs (π*_equatorial_). Although π_equatorial_ MOs are all involved in the [Re≡N]^2+^ multiple bond, π*_equatorial_ can easily overlap with filled orbitals of donor ligands spanning the two residual positions on the trigonal plane. The non-bonding filled FO (*n*) can bind with π-acceptor ligands along the axis perpendicular to the trigonal plane (Figure 6). This interaction nicely explains the tendency of the π-acceptor tertiary phosphine PPh_3_ to ideally occupy the two axial positions of a pseudo *tbp* geometry in the rhenium nitrido complex [ReNCl_2_(PPh_3_)_2_] [33]. As a result, the [Re≡N]^2+^ functional group in a *tbp* geometry exhibits a unique and elegant chemistry depending on the type of coordinating ligands. With donor ligands, it can behave as a Lewis acid on the equatorial trigonal plane, whereas it acts as a Lewis base along the axial positions. Notably, the same structural behaviour was observed with the [Tc≡N]^2+^ functional group [34,35] and, by exploiting these peculiar chemical properties, it was possible to develop an unprecedented class of heteroleptic ^99m^Tc complexes showing high myocardial uptake and favourable biodistribution characteristics [35,36,37,38,39,40,41,42,43,44].

## 5. Chemical Differences between the Re(N) and Re(O) Cores

Despite the similarity between the FOs diagrams of Re(N) and Re(O) functional groups, there exist marked differences in their chemical behaviour. For example, the remarkable property of complexes containing the Re(V) nitrido group to switch between *sp* and *tbp* geometries upon coordination by π-acceptor ligands is not displayed by Re(V) oxo complexes. Conversely, the binding of π-acceptor ligands to the [Re≡O]^3+^ moiety invariably forces the coordination arrangement to turn into octahedral geometry [45]. This geometrical change from square-planar to octahedral coordination may trigger the formation of the Re(V) dioxo group, *trans*-[O = Re = O]^+^, where a second oxygen atom binds the [Re≡O]^3+^ group in *trans* position to the oxo atom. Usually, the second oxygen atom originates from a doubly deprotonated water molecule after its nucleophilic attack to the Re(V) centre. If the *trans*-[O = Re = O]^+^ group is not sufficiently stabilized by some appropriate ligand, this event may initiate the subsequent chain of hydrolytic attacks to the Re(V) ion, which finally leads to the formation of the perrhenate anion [ReO_4_]^−^. Ligands capable of stabilizing the *trans*-[O = Re = O]^+^ core are bidentate diphosphines [R(R′)–P–P–(R′)R where R, R′ = organic substituents], ethylenediamine, and acyclic and cyclic tetramines. Figure 7 illustrates a few examples of octahedral complexes comprising the *trans*-[O=Re=O]^+^ moiety [6,7,8,22,25,46].

The chemical factors that are usually invoked to explain the chemical differences between the [Re≡N]^2+^ and [Re≡O]^3+^ functional groups are the lower electronegativity and smaller atomic radius of the nitrogen atom with respect to the oxygen atom. This allows a more extensive overlap of the valence AOs of Re and N, and the formation of a strong Re≡N multiple bond [48]. Notably, in many Re(V) oxo octahedral complexes, the Re(V) oxo bond distance is closer to that of a Re=O double bond than to that of a Re≡O triple bond predicted by the FOs diagram reported in Figure 3.

In *sp* Re(V)-nitrido complexes, the shorter Re≡N bond length determines a strong distortion of their square pyramidal geometry that generate their characteristic umbrella-shaped structure. This geometrical distortion causes the metal atom to move up from the square plane towards the nitride nitrogen atom with the concomitant bending of the four coordination bonds. This effect can efficiently shield the Re(V) atom from the nucleophilic attack of an incoming water molecule in the *trans* position of the nitrido nitrogen atom. Therefore, this geometrical distortion can reasonably explain the higher inertness of Re(V) nitrido *sp* complexes against hydrolysis by water molecules [18,33,48,49,50,51]. One example of Re(V) nitrido complexes with a pseudo *tbp* geometry is illustrated in Figure 8a, while Figure 8b shows the structure of a bis-substituted dithiocarbamate complex with a distorted *sp* geometry. It is worth noting here that, for the sake of simplicity, in Figure 8a, *D_3h_* symmetry has been ideally (and arbitrarily) attributed to the complex [Re(N)Cl_2_(PPh_3_)_2_] whereas its real geometry is distorted by the sterically encumbering PPh_3_ substituents and better described by a *C_2v_* symmetry.

### Ligands for the Re≡N and Re≡O Functional Groups

The diagrams illustrated in Figure 3 and Figure 5 can be also conveniently used to infer which type of ligands are the most appropriate for binding to the Re≡N and Re≡O functional groups. In *sp* geometry, the FOs of both rhenium cores are formed by two degenerate, high-energy, empty π*-types orbitals and, consequently, these two cores behave as soft Lewis acids. Accordingly, they react preferentially with electron-rich Lewis bases able to span the four coordination positions on the square planar plane. Therefore, chelating ligands containing negatively charged, closed-shell donor atoms such as oxygen, sulphur, and mono-deprotonated nitrogen, are particularly appropriate for binding the Re(V) nitrido and Re(V) oxo cores in a five-coordinated, square pyramidal arrangement. Figure 9 illustrates the general structures of bidentate, tridentate, and tetradentate ligands that have been used for the high-yield production of stable ^188^Re radiopharmaceuticals. These ligands are characterized by the presence of electron-rich soft donor atoms and, with the [Re≡O]^2+^ and [Re≡N]^2+^ groups, commonly they form mono- and bis-substituted, distorted *sp* complexes where the Re(X) group (X = O, N) occupies an apical position and the ligands span the four coordination positions on the square plane.

## 6. Matching the Nuclear and Biological Properties of ^188^Re Radiopharmaceuticals

Current applications of ^188^Re in the clinic have been reviewed extensively in other articles [1,3,4,25,26,27]. This review aims to discuss a more basic approach for a rational design of ^188^Re radiopharmaceuticals by considering both nuclear and electronic properties of rhenium complexes.

To translate ^188^Re as a therapeutic radionuclide to the clinic, a robust production method needs to be in place. Inspired by the chemical similarities between rhenium and technetium, many research groups attempted to use the kit formulations developed for the preparation of ^99m^Tc radiopharmaceuticals to produce the corresponding ^188^Re radiopharmaceuticals. However, when studied at the tracer level, the predicted chemical similarities were found to be more elusive than expected. As mentioned previously, only a small number of ^188^Re radiopharmaceuticals have been successfully introduced into the clinical use using freeze-dried kit formulations. The design of these radiopharmaceuticals was almost invariably based on the chemistry of Re(V) oxo and Re(V) nitrido cores and of ^188^Re complexes containing the sulphur rich Re^III^(S_3_)_2_(S_2_) metallic group [3,4,18,25,27,28].

Indeed, it is becoming increasingly important to obtain an adequate match between the nuclear characteristics of the radionuclide, the pharmaceutical carrier, and the proposed biological target to elicit the most powerful therapeutic effect on the disease profile. This strategy can also favour a higher throughput from development to clinical application. The high-energy and relatively short-lived β^−^ emission of ^188^Re suggests that high-capacity target sites could be more suitable for therapeutic applications based on this radionuclide. Current research on targeted radionuclide therapy is mostly focused on selecting interactions of the therapeutic agent with biological substrates expressing low-capacity receptors. These sites should be more appropriate for radionuclides emitting particles with high linear energy transfer (LET). This category includes α (e.g., terbium-149, astatine-211) and Auger emitters (e.g., iodine-125, copper-64). It is worth noting that a critical issue related to these receptor-specific therapeutic radiopharmaceuticals is that their uptake should always occur in proximity to the cell nucleus to ignite the disruption of the cancerous cell. For this approach, therefore, a strong binding to the receptor followed by an efficient internalization of the radiolabelled vector are necessary prerequisites. Conversely, rather than focusing on receptor interactions and internalization, ^188^Re-based radiopharmaceuticals should be designed to target high-capacity sites that commonly allow a rapid and extensive accumulation of the cytotoxic payload at the tumour site associated with a fast clearance from background tissues. To strengthen this statement, a few examples of clinical applications of ^188^Re radiopharmaceuticals designed to adhere to these principles are highlighted and summarized below.

An illustrative example of the obstacles that were encountered in attempting to apply ^99m^Tc freeze-dried kit formulations to ^188^Re is offered by the preparation of ^188^Re bisphosphonates for bone pain palliation. The extemporaneous method used to produce ^99m^Tc bisphosphonates involves the simple mixing of a lyophilized powder containing SnCl_2_, a selected bisphosphonate ligand, and other excipients with generator-eluted [^99m^Tc][TcO_4_]. This labelling reaction generates a mixture of more than 14 different species of ^99m^Tc-complexes of unknown structure. Only one of these complexes have been isolated and structurally characterized [52], but experimental results clearly showed that these species were polymeric in nature where bisphosphonate ligands bridged several Tc and Sn atoms to form a multinuclear complex. Most importantly, the polymeric structure was recognized to be essential for ensuring the efficient bone accumulation of these agents.

Surprisingly, when this kit formulation was utilized for producing the corresponding ^188^Re bisphosphonates, bone uptake was not observed. The trivial reason was that, as discussed in Section 2, the low standard reduction potential of [ReO_4_]^−^ was not sufficient to allow the efficient reduction of the Re(VII) metal centre and the formation of a ligand-bridged polymeric network with Sn^2+^ ions. This effect is more pronounced at a non-carrier added level and to increase the reduction potential and the probability of formation of bisphosphonate bridges, cold rhenium needs to be added to the kit formulation. For this reason, reactor-produced ^186^Re is often chosen as an alternative, since this production method usually yields a carrier added radioisotope. The ^188^Re freeze-dried kit commonly contains 3 vials. The first vial contains cold ammonium perrhenate solution used to increase the carrier rhenium in the final radiopharmaceutical for a more optimal bone uptake. The second vial contains the selected bisphosphonate ligand (most often, HEDP = 1-hydroxyethylidene-1,1-diphosphonic acid), stannous chloride, and gentisic acid. The third vial simply contains a sodium acetate buffer to ensure that the pH of the final solution is in the physiological range. The multiple-vial kit demonstrated very robust radiolabelling yields and radiochemical purities above 95% [29,53]. Although the amount of cold rhenium carrier is critical to obtain a radiopharmaceutical with the necessary radiochemical purity as well as the physiological bone uptake needed to elicit the therapeutic response [54], this result was achieved at the cost of a lower radionuclidic purity that, obviously, decreased the effective dose deposited on the bone lesions. Bisphosphonates labelled with ^188^Re [55] for bone pain palliation have demonstrated impressive results in clinical studies. Since painful bone metastases are treated with radionuclide therapy at an advanced stage, the target site has a high capacity with widespread metastases. Near ideal pharmacokinetics results in a quick clearance, nearly 40% excreted within 8 h post-injection via the kidneys. A high percentage is localized in the target, 40% accumulated in the skeleton at 24 h post-injection. Optimally, a prolonged half-life in tumour (269 ± 166 h) is reported with a short biological half-life (51 ± 43 h) for the unaccumulated radiopharmaceutical. Therapy with [^188^Re]Re(HEDP) has demonstrated an improvement in overall survival and progression-free survival in patients with bone metastases and even a reduction in prostate-specific antigen (PSA) levels in metastatic castration-resistant prostate cancer patients. Therefore, ^188^Re-based therapy might elicit an additional antitumor effect not restricted to pain palliation [30,53,54,55,56,57,58,59].

By application of the chemistry of the Re(N) core as described in Section 4, a robust two-vial kit formulation has been proposed for the successful labelling of the iodinated poppy oil lipiodol. This oil is used as contrast agent for X-ray liver imaging. The first vial is used to produce the intermediate [^188^Re][Re≡N]^2+^ functional group. The second vial contains the ligand DEDC (diethyldithiocarbamate = DEDC) to form the final [^188^Re]ReN(DEDC)_2_ complex that is subsequently extracted into the hydrophobic lipiodol phase [18]. The initial formulation of this kit also included in the first vial a glacial acetic acid and sodium oxalate mixture. Recently, a simplified and more advanced labelling procedure has been described where this mixture is replaced by 0.5 M oxalate buffer [23]. Extraction of [^188^Re]ReN(DEDC)_2_ into the organic phase affords the final [^188^Re]Re-lipiodol. This agent is then selectively administered and deposited into a liver tumour through a catheter inserted into the hepatic artery and used for the treatment of hepatocellular carcinoma. Therefore, its therapeutic action occurs through a nonspecific and non-saturable targeting mechanism, which does not rely on advanced molecular targeting, but rather on the chemo-absorption of a highly lipophilic substance by cancerous cells. This method offers an inexpensive treatment for hepatocellular carcinoma and demonstrates minimum off-target radiation with no mechanism of excretion needed [60,61].

Similarly, the local application of ^188^Re skin cream for brachytherapy of non-melanoma skin cancer (NMSC) [62,63] also relies on physical methods of localization and does not involve systemic administration and localization through a vector at the target site. With both therapies, positive clinical results and no severe complications are reported. Again, these therapies do not rely on sophisticated interactions with saturable/limited receptor targets.

Other mechanisms of accumulation of ^188^Re-based therapies worth mentioning are the use of rhenium-188 as a substrate for the sodium/iodine symporter (NIS) and the development of a tumour microenvironment targeting radiopharmaceuticals. Since the perrhenate anion ([^188^Re][ReO_4_]^−^) exhibits the same biological behaviour as pertechnetate ([^99m^Tc][TcO_4_]^−^) and iodide ([^131^I][I]^−^), it is an attractive alternative for the treatment of tumours that expresses the sodium/iodide symporter. Current investigations not only focus on the thyroid malignancies, but also on human breast cancer [64] or even other malignancies such as cervical cancer transfected express the NIS symporter [65]. An in-depth review of NIS-based technologies for radionuclide imaging and treatment is provided in a recent summary [66]. Finally worth mentioning is the more recent development of radiopharmaceuticals that target the tumour microenvironment in particular cancer-associated fibroblasts (CAFs) expressing fibroblast-activating protein (FAP), tumour-associated macrophages (TAMs), and tumour angiogenesis. By using the tumour microenvironment as a target, a more robust and larger volume delivery platform is made available that might be less prone to downregulation and the development of resistance. Although only ^99m^Tc-radiopharmaceuticals were presented, the use of ^188^Re as a therapeutic-matched pair for FAP targeting ligands was proposed by Lindner and co-workers [67].

## 7. Conclusions

The last decade has seen a growing trend towards targeted receptor radionuclide therapy of low-capacity receptor sides in elegantly designed complex systems. Hence, our proposal that high-capacity receptor sites should be revisited and might even be more beneficial than low-capacity receptor sites for cancer treatment might seem controversial.

As results of larger multi-center clinical trials on therapeutic agents such as Luthathera^®^ and Pluvicto^®^ [designed to selectively target tumors overexpressing somatostatin receptors or prostate specific membrane antigen (PSMA), respectively] become available, they are expanding our understanding of radionuclide therapy [68,69]. It is now well established that these agents prolong overall survival in patients affected by neuroendocrine tumors or prostate carcinoma; unfortunately, they fail to completely eradicate the disease. In some patients, radio resistance develops quite early in the therapy. Serious doubts regarding the theoretical foundations on which the therapy based on the targeting of low-capacity receptors was built is now being raised. Phenomena such as the genetic instability of malignant cells and the downregulation of receptor expression caused by the same targeted radionuclide therapy have been suggested as responsible for the partially unsatisfactory clinical results observed in patients [11,70,71]. For these reasons, it might be advantageous to shift attention to non-saturable biological substrates capable of absorbing a larger cytotoxic radioactive payload. It is proposed that these targets would be less affected by genetic instability and downregulation. Therefore, for these larger sites, it is necessary to use radionuclides which emit more penetrating, high-energy particles with a relatively shorter half-life to allow more frequent administrations of the therapeutic agent. In this context, it is reasonable to assume that ^188^Re could play an important role given its favorable nuclear characteristics. It is also noteworthy that, since ^188^Re is produced by a generator, this has made it much easier to develop automatic modules to produce ^188^Re radiopharmaceuticals as already documented in the scientific literature [61,72,73]. Automation can strongly facilitate the preparation of ^188^Re radiopharmaceuticals under strict GMP conditions and drastically reduce operator exposure. The decentralized production of therapeutic radiopharmaceuticals can be a game changer in nuclear medicine, enhancing theranostics to the same extent as occurred in diagnostic imaging by the introduction of the ^99^Mo/^99m^Tc generator. This goal is well worth the combined efforts of the nuclear medicine community.

## Figures and Tables

**Figure 1 molecules-28-01487-f001:**
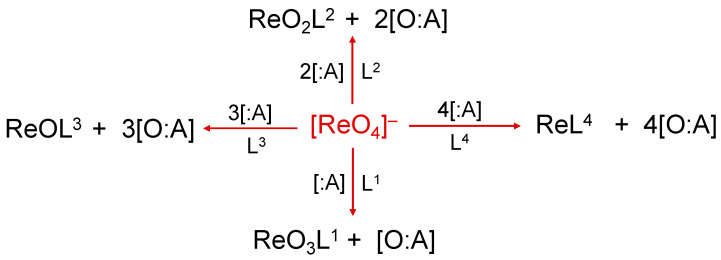
Oxygen atom transfer reactions using [^188^Re]perrhenate as starting substrate. The symbol ‘:A’ represents the oxygen-atom acceptor and L*^n^* (*n* = 1–4) the ligand (or multiple ligands) suitable to form a stable complex with one of the various intermediate Re-oxo forms. Examples of :A are SnCl_2_ or tertiary phosphines.

**Figure 2 molecules-28-01487-f002:**
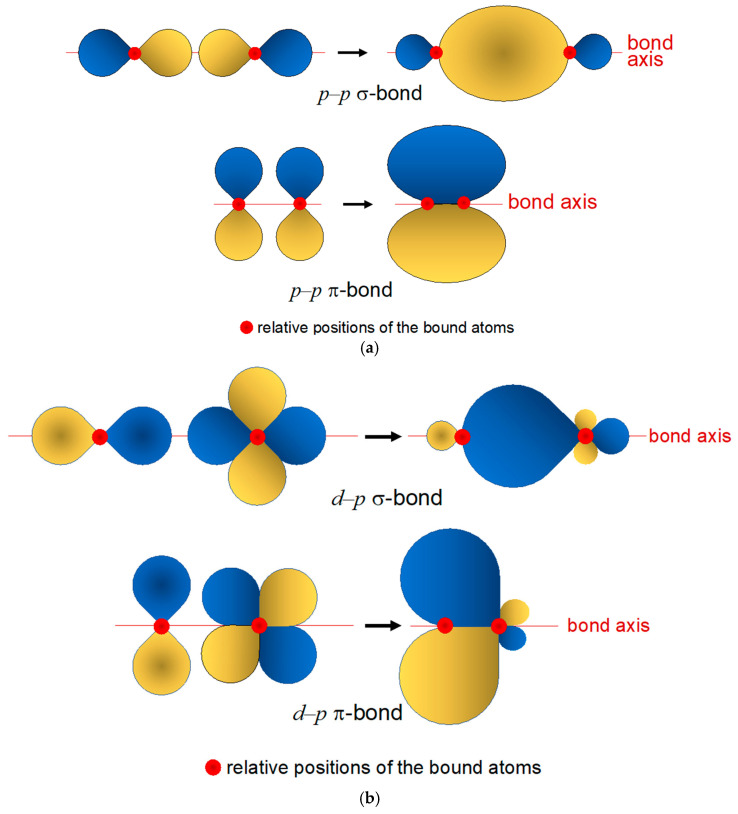
Examples of formation of σ and bonding orbitals from the overlap of (**a**) *p*–*p* and (**b**) *d*–*p* orbitals.

**Figure 3 molecules-28-01487-f003:**
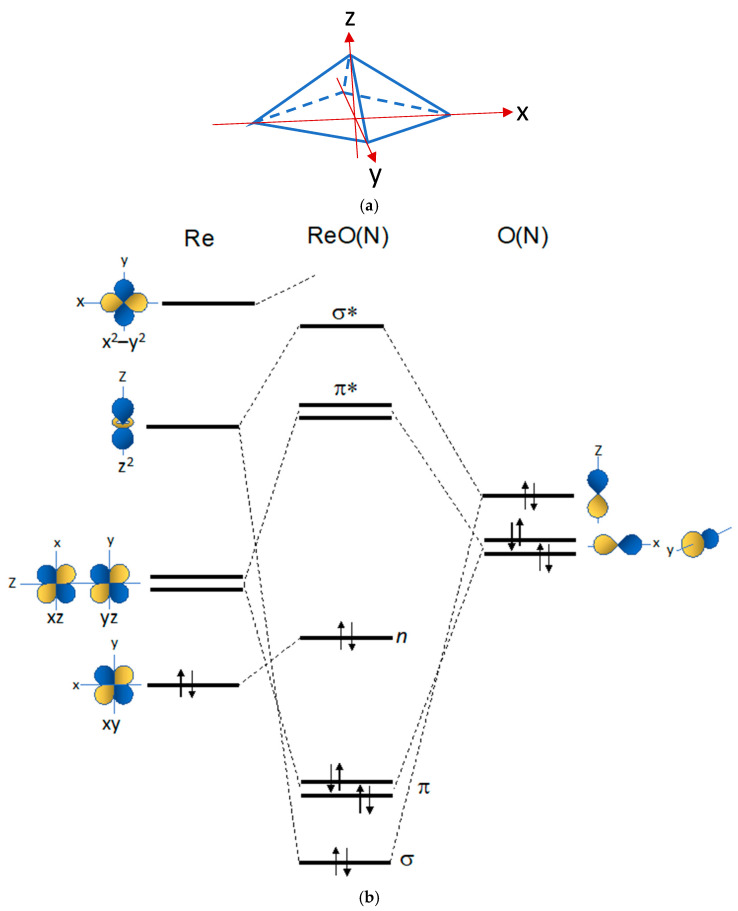
(**a**) Square pyramidal geometry (*sp*). In this geometry, the rhenium atom lies at the centre of the basal square-plane, and the heteroatom (O or N) spans the apical position of the square pyramid. (**b**) Qualitative energy level diagram of frontier molecular orbitals (FOs) for the [Re≡N]^2+^ and [Re≡O]^3+^ inorganic functional groups in a *sp* geometry (energy is reported on the vertical *y*-axis).

**Figure 4 molecules-28-01487-f004:**
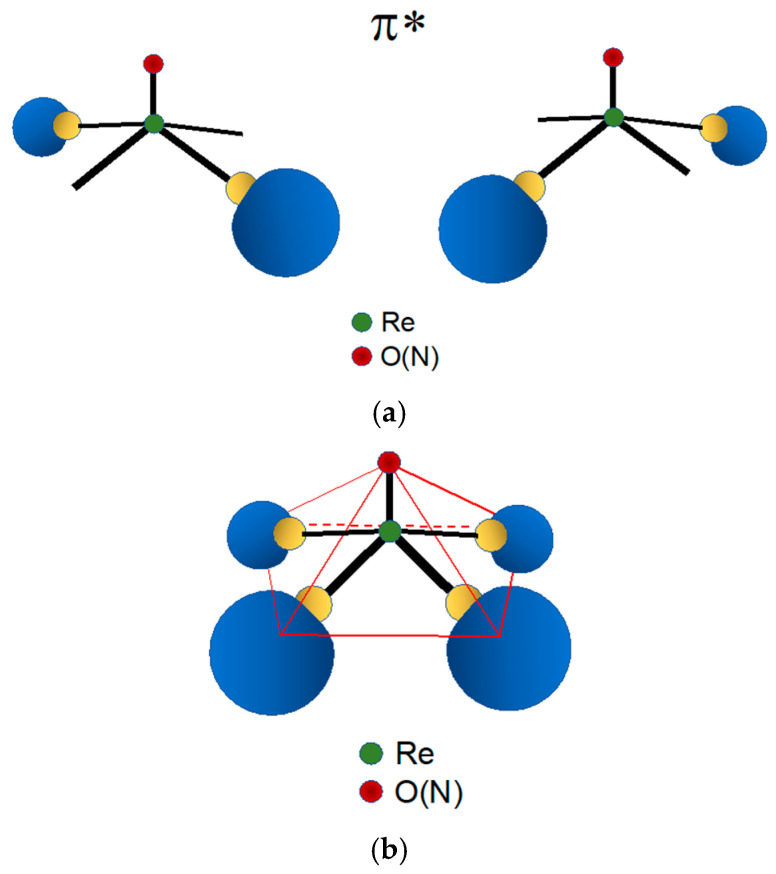
(**a**) Qualitative probability density distribution of the two empties, isoenergetic π* frontier orbitals (FOs) of the Re(X) (X = O, N) functional group in *sp* symmetry. (**b**) Arrangement of the two FOs in a square pyramidal geometry. The highest density is distributed along the lines connecting the central metal to the coordinating atoms located on the four positions on the basal square plane. It is important to note that in the figure the lobes of the MOs were arbitrarily displaced along the bond axis and not centred on the metal atom. This unconventional representation was preferred to highlight how the spatial arrangement of the electronic density nicely accounts for the square pyramidal geometry of the resulting complexes.

**Figure 5 molecules-28-01487-f005:**
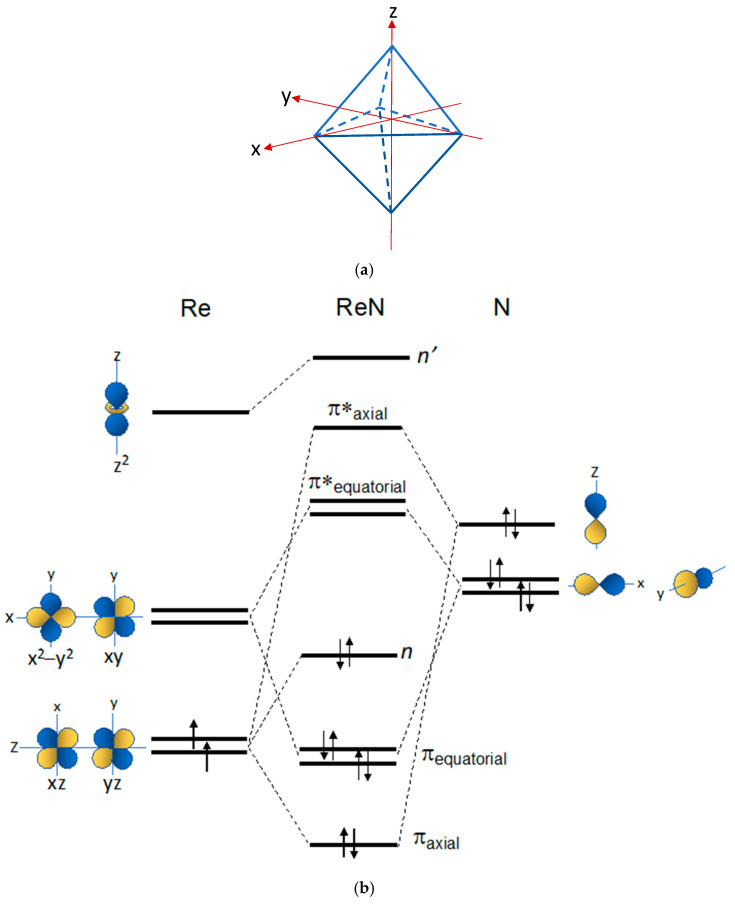
(**a**) Trigonal-bipyramidal geometry (*tbp*). In this geometry, the rhenium atom lies at the centre of the equatorial trigonal plane and the heteroatom (N) spans one vertex on the same plane. (**b**) Qualitative energy level diagram of frontier molecular orbitals (FOs) for the [Re≡N]^2+^ group in a *tbp* geometry (energy is reported on the vertical y-axis). In the Figure, *n* and *n’* are non-bonding FOs.

**Figure 6 molecules-28-01487-f006:**
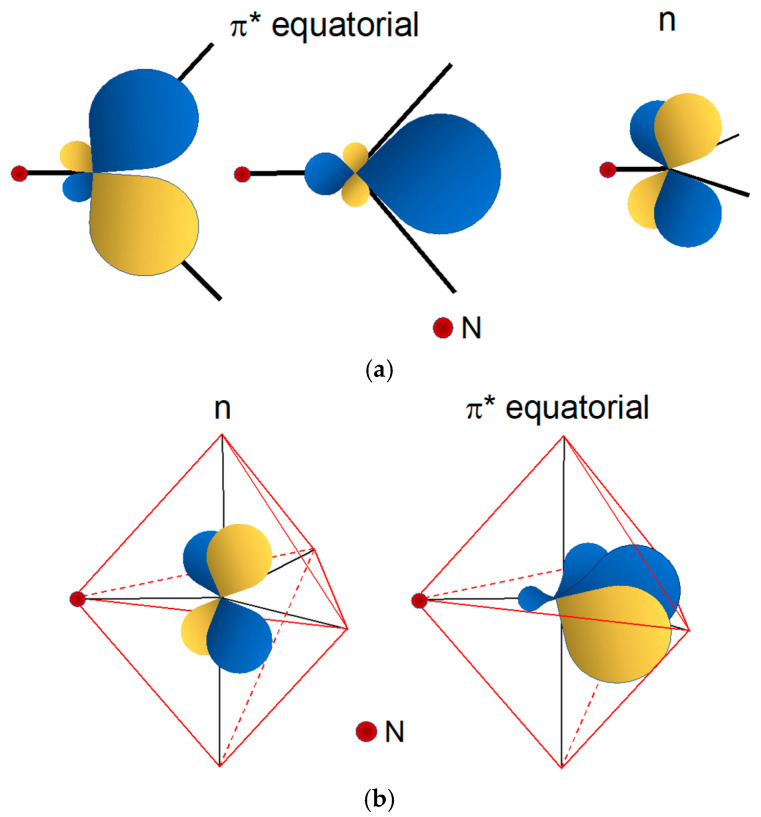
(**a**) Qualitative probability density distribution of the two isoenergetic π* frontier orbitals (FOs) and of the non-bonding *n* orbital of the Re(N) functional group in a *tbp* symmetry. (**b**) Arrangement of the three FOs in a trigonal bipyramidal geometry. The two empties π* FOs lie on the trigonal equatorial plane and are separated from the filled *n* FO positioned on a perpendicular plane.

**Figure 7 molecules-28-01487-f007:**
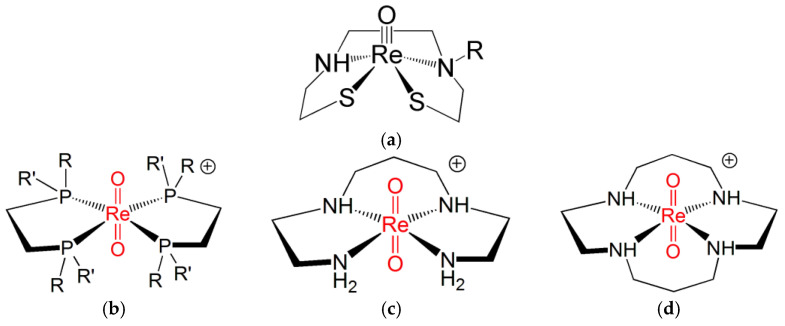
The molecular structures of (**a**) an example of *sp* Re(V) oxo complex and of octahedral *trans*)-dioxo Re(V) complexes with (**b**) diphosphines and (**c**) acyclic and (**d**) cyclic tetramines ligands (R, R′ = any pendant organic substituents) [6,7,8,25,46,47].

**Figure 8 molecules-28-01487-f008:**
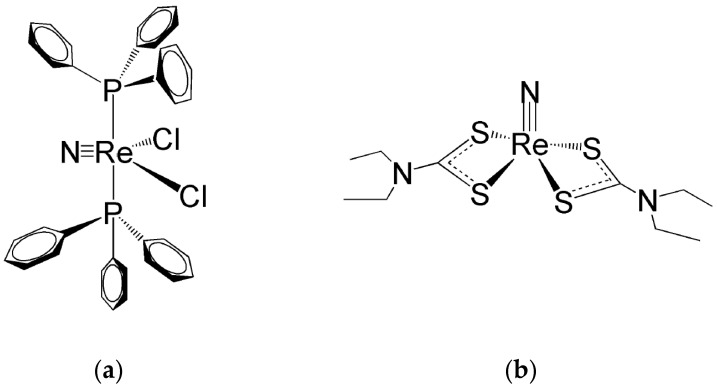
(**a**) The mixed halogeno-phosphino Re(V) complex (Re(N)Cl_2_(PPh_3_)_2_) [33] and (**b**) the bis-substituted complex (Re(N)(CS_2_NEt_2_)_2_) [51].

**Figure 9 molecules-28-01487-f009:**
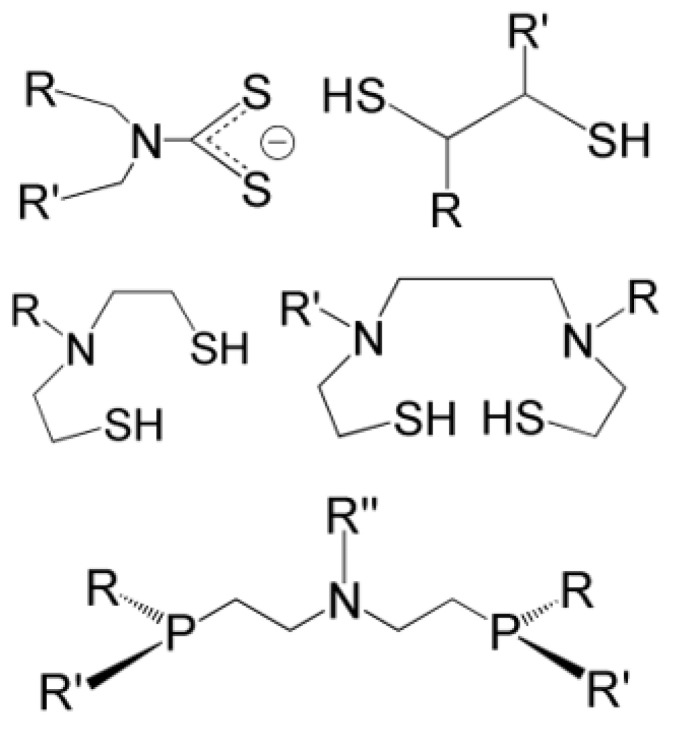
The general structure of some examples of bidentate and tetradentate ligands employed in the production of monoxo and nitrido ^188^Re radiopharmaceuticals with *sp* geometry ((R, R’, R” = any pendant organic substituent) [6,8,25,49,50].

## Data Availability

Not applicable.

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
