# Peer review of "Fundamentals of Rhenium-188 Radiopharmaceutical Chemistry"

_molecules, 2023, doi:10.3390/molecules28031487_

Round 1
Reviewer 1 Report
The manuscript submitted by Kleynhans, Duatti and Bolzati addresses an interesting and important field of nuclearmedical chemistry and the title and the abstract promise valuable information about the "Chemistry" of 188Re complexes and their current and prospective use in nuclear medical chemistry. But unfortunately, not all the information related with these "promises" are delivered. On the contrary, some of the 'fundamental' discussions included in the manuscript may produce confusion with the readers rather than clarity (and is superfluous in the context of the title of the present manucript: "188Re radiopharmaceuticals"). On the other hand, some recent research about potential 188Re/99mTc pairs are not mentioned at all. Thus, in my opinion the submitted manuscript deserves a careful rewriting with a clearer focus to the radiopharmaceutical aspects.
Following major aspects should be considered:
(1) The entire Chapter 2 should be rewritten (or better completely omitted). The general discussion about chemical concepts such as 'oxidation states' has less to do with the central matter of the present mnuscript. Historical sketches ("a planet covered by water') and the focus of the redox chemistry to 'oxygen transfer chemistry' as done in Chapter2 and particularly Figure 1 is confusing and not (no more) appropriate. We all know that most of the fundamental concepts (as 'redox') are formal ones. But everything is easy to explain by electron transfer (or localization in appropriate molecular orbitals) in the compounds and between the compounds involved in redox reactions. This also may help to follow the arguments of redox potential presented later. Nearby, the authors permanently use the 'formal' oxidation states, when they claim Re(V) and Re(VII) compounds.
(2) A second major concern is related to the long discussion of aspects of the chemical bonding in oxido and nitrido complexes of rhenium. The extended discussion of the structural chemistry (square pyramidal vs. trigonal bipyramidal) is more or less useless as long it is done exclusively on a qualitative level. As far as I remember, tpb structures are mainly found when two bulky ligands such as PPh3 are present and occupy the axial positions in such complexes. Thus, steric factors should also play a role, which should at least be checked by simple DFT calculations (e.g between [ReNCl2(PPh3)2] and the hypothetical compound [ReNCl2(PMe3)2]) . I suppose that the enery difference between the borderline structures will be small, also having in mind that six-coordinate complexes are formed, when only somewhat smaller phosphines than PPh3 are used.
Some more minor aspects, which should be considered during the revision.
(1) Please use a consistent format to write isotopes. I suggest the IUPAC notation '188Re'. Currently, 'rhenium-188', '188Re', '188Re', '[188Re][ReO4|', 'Tc-99m' etc.
(2) There is an extensive use of hyphens throughout the manuscript, e.g. 'Re-compounds', Re(V)-oxo, Re(V)-nitrido, 188Re-radiopharmaceuticals etc. This is not common in English.
(3) page 2, line 48: technetium is a 'second row', rhenium is a 'third row' transition metal.
(4) page 2, line 56: According to IUPAC technetium and rhenium belong to 'Group 7', not 'Group VII B'.
(5) page 3, line 102: A strange and confusing way to write the formula of sodium perrhenate. Please correct.
(6) page 6, line 236 and following: The discussion about the claimed oxalato complex of rhenium(VII) should be supported by experimental data or an appropriate reference with an experimental proof. Otherwise it sounds like speculation.
(7) page 12, line 434: Probably 'Figure 7' is meant.
(8) page 13, line 451: The discussed difference in the bond lengths (161 pm vs. 165 pm is not at all significant.
(9) page 13, line 458: 'Square planar' or 'square pyramidal' ???
(10) page 14, line 480: 'five-coordinate' instead of 'five-coordinated'
(11) Chapter 6: Some Figures containing the chemical formulae of the compounds should be included for a better understanding (even when they are partially shown before).
(12) Figures 7, 8 and 9: References should be included to show the sources, where the compound have been taken from.
(13) References: 18 out of 67 references are self-citations. The number is not a problem having in mind the major contributions the authors have delivered to this field of research in the past. But many other major contributions (e.g. the sources of the compounds discussed, some of the chemical reactions discussed) are not cited. Please add an adequate number of references to cover the entire field.
Reviewer 2 Report
This is a useful review article that will be of interest to workers in the field of radiopharmaceuticals in particlular as well as transition metal chemists in general. I recommend acceptance after several major points are addressed. Points of major concern to me are:
1. The molecular orbital analysis of the ReN tbp core is based upon a D3h point group. The description provided of the core, however, puts the nitride ligand in the equatorial plane which gives the complexes a C2v point group ( given the ligands presented) or at best a C4v point group if all of the remaining ligands were identical. The degenerate pairs of MOs reported are not degenerate in C2v symmetry. The section on the ReN MO structure should address the correct point group for the core.
2. The section on oxidation states (~line 126 through line 149) seems to bristle against the useful concept of oxidation states which are, in turn, based upon the concept of electronegativity. Later the paragraph which runs from lines 177-187 quite sensibly adoptes the concept of oxidation and reduction which is, of course, based upon the concept of oxidation states. Clearly a key difference in the preparation of radiopharmaceuticals from the pertechnate ion and the perrhenate ion is the commonly occurring smaller reduction potential for higher oxidation states of the heaviest congeners of the transition metals. This might be more easily expressed as Eocall = Eocathode - Eoanode with a note about the smaller Eocathode for ReO4- versus TcO4-.
3. In the paragraph that runs from line 210 through line 219, the change in coordination number at the rhenium center is claimed to be the entire system for the Gibbs Equation (Delta G = Delta H - T Delta S). Since the complex is not the entire component of the reaction, the change of entropy of all reactants and products combined must be considered with respect to thermodynamic considerations.
4. In Figure 2, the nuclei locations should be indicated. In illustration (b) the d-p sigma bond is mislabelled as a pi bond. Also in (b), a yellow lobe is missing for the sigma bond. The shape of the pi bond MO in (b) will probably need to change when the nuclei are indicate. Directions (x, y, and /or z) should be included in Figure 2 and the exact orbitals being depicted should be labelled.
5. In Figure 3 (b) symmetry labels labels should be used for all orbitals (AO and MO). C4v is a better descriptor than sp for square pyramidal. The s orbital and its participation in the sigma bond for O ro N should be mentioned.. The Re d orbitals should be presented as degenerate on the left side of the diagram for they only become degenerate upon interaction with the ligands. The same is true for the set of p orbitals for N or O. It should be explained, in some fashion, why dx2-y2 is at higher energy. In my opinion, the diagram should present not just the "core" but the entire set of orbitals in order to make sense for the novice reader.
6. In Figure 4, the authors have changed their convention of blue and yellow from Figures 2 and 3. In Figures 2 and 3, blue and yellow indicate opposite phases of the underlying wave. In Figure 4, yellow seems to represent metal-based orbitals and blue seems to indicate ligand-based orbitals with no consideration of the underlying phase of the wave. The conventions of Figures 2 and 3 should be used for Figure 4 or the change in conventions should be made explicit.
Minor concerns.
1. Examples of :A should be given and consideration should be presented of anionic ligands in Figure 1.
2. The paragraph that runs from line 146 through line 158 which discusses valency seems unnecessary given the more modern definitions of valency.
3. Since the Nernst Equation is discussed it should probably be presented.
4. Delta Eo (line 191) is really Eocell.
5. line 26 frontiers
6. line 33 applications
7. line 38 outperforms
8. line 45 188 should be superscript
9. line 47-49 Tc is a member of the 2nd row transition metals
10. Group 7 (IUPAC) lines 49 and 56
11. line 51 the use of investigated implies there should be a reference
12. line 78 constraints
13. line 97 fortuitous
14. line 167 two thats
15. line 264 two only
16. For the paragraph that runs lines 263-272, a scheme for the prep of other radiopharmaceuticals would be good given that this is a review article.
17. metallic rhenium should become rhenium atoms (line 278)
18. All Lewis base ligands are sigma donors (line 286)
19. line 286 There are pi donor ligands and pi acceptor ligands. Pi donors tend to stabilize high oxidation states and pi acceptor ligands tend to stabilize lox oxidation states.
20. line 345 the hence needs more explanation.
21. The Figure 4 caption has empties and probably means the empty orbitals?
22. The paragraph that runs from line 366 to 376 is unclear.
23. References should be indicated for the reactions in lines 369-372.
24. The paragraph than runs line 499-508 and 530-539 are identical.
25. line 553 - What is paragraph 2.0?
26. line 554 of needs to be added
27.. line 589 glacial acetic acid
28. 0.5 M (line 591)
29. line 609 exhibits
30. line 611 express
31. The font changes in the Conclusions section.
Round 2
Reviewer 1 Report
The submission has improved due to the corrections of the issues I have summarized in my 'minor corrections list'. My major concerns with the content of the paper, however, have not been addressed at all or at least not satisfactory. The authors disagreed with almost all my objections.This means that they want to have the paper in the unmodified form.
This is O.K. to me, since I am just a reviewer. When the Editors feel that it should be published in this way (and the author Cristina Bolzati is the scientific Editor of this issue) then it is perfectly fine to me.
Thus, please understand my recommendation 'Accept in present form' in this way and not as a strong support for the content of the paper in its present form.
Reviewer 2 Report
The topic will be of interest to many and the manuscript is generally well done.
One major concern is the qualitative MO descriptions and their applications. The authors describe qualitative MO diagrams for square pyramidal (sp) ReO cores and for trigonal bipyramidal (tbp) ReN cores. The complex characterized in Reference 33 is not tbp. The point group for the complex in Reference 33 is C2v. The manuscript loses some credibility in seeming to force a tbp geometry onto the complex in order to discuss differences in reactivity.
I would recommend the removal of the Mo diagrams and the discussion but I also believe that the authors should be able to present their work as they see fit.
Figure 4 seems to adopt a different formalism for shading lobes of orbitals than Figures 2, 3, and 5. A consistent approach would be less confusing.
Minor points:
Line 16 118Re should be 188Re
188Re[ReO4-] and 188Reperrhenate are both used to represent the same species. One form or the other might be adopted consistently.
Line 26 "frontiers" should be "frontier"
At the top of Page 6, the authors should note that the following reaction does not constitute an increase in entropy for the universe:
L + ReOL4 --> ReOL5
The authors' description makes it seems like the above reaction is favorable with respect to entropy.
Line 367 seems like it intends [O]2- and [S]2- rather than [O]- and [S]-
